# Diagnosis of latent tuberculosis infection is associated with reduced HIV viral load and lower risk for opportunistic infections in people living with HIV

**Katharina Kusejko**[1,2]*, **Huldrych F. Günthard**[1,2], **Gregory S. Olson**[3], **Kyra Zens**[4], **Katharine Darling**[5], **Nina Khanna**[6], **Hansjakob Furrer**[7], **Pauline Vetter**[8], **Enos Bernasconi**[9], **Pietro Vernazza**[10], **Matthias Hoffmann**[11], **Roger D. Kouyos**[1,2◑], **Johannes Nemeth**[1◑]*, **the Swiss HIV Cohort Study**[¶]

1 Division of Infectious Diseases and Hospital Epidemiology, University Hospital Zurich, Zurich, Switzerland, 2 Institute of Medical Virology, University of Zurich, Zurich, Switzerland, 3 Seattle Children's Research Institute, Seattle, Washington, United States of America, 4 Institute of Experimental Immunology, University of Zurich, Zurich, Switzerland, 5 Department of Infectious Diseases, University Hospital Lausanne, Lausanne, Switzerland, 6 Infectious Diseases and Hospital Epidemiology, University Hospital Basel, Basel, Switzerland, 7 Department of Infectious Diseases, Bern University Hospital, University of Bern, Bern, Switzerland, 8 Division of Infectious Diseases, Geneva University Hospitals, Geneva, Switzerland, 9 Division of Infectious Diseases, Regional Hospital Lugano, Lugano, Switzerland, 10 Division of Infectious Diseases, Cantonal Hospital St Gallen, St. Gallen, Switzerland, 11 Division of Infectious Diseases, Cantonal Hospital Olten, Olten, Switzerland

◑ These authors contributed equally to this work.
¶ Membership of the the Swiss HIV Cohort Study is listed in the Acknowledgments.
* katharina.kusejko@usz.ch (KK); Johannes.nemeth@usz.ch (JN)

**Data Availability Statement:** The individual level datasets generated or analyzed during the current

## Abstract

Approximately 28% of the human population have been exposed to *Mycobacterium tuberculosis* (MTB), with the overwhelming majority of infected individuals not developing disease (latent TB infection (LTBI)). While it is known that uncontrolled HIV infection is a major risk factor for the development of TB, the effect of underlying LTBI on HIV disease progression is less well characterized, in part because longitudinal data are lacking. We sorted all participants of the Swiss HIV Cohort Study (SHCS) with at least 1 documented MTB test into one of the 3 groups: MTB uninfected, LTBI, or active TB. To detect differences in the HIV set point viral load (SPVL), linear regression was used; the frequency of the most common opportunistic infections (OIs) in the SHCS between MTB uninfected patients, patients with LTBI, and patients with active TB were compared using logistic regression and time-to-event analyses. In adjusted models, we corrected for baseline demographic characteristics, i.e., HIV transmission risk group and gender, geographic region, year of HIV diagnosis, and CD4 nadir. A total of 13,943 SHCS patients had at least 1 MTB test documented, of whom 840 (6.0%) had LTBI and 770 (5.5%) developed active TB. Compared to MTB uninfected patients, LTBI was associated with a 0.24 decreased log HIV SPVL in the adjusted model (*p* < 0.0001). Patients with LTBI had lower odds of having candida stomatitis (adjusted odds ratio (OR) = 0.68, *p* = 0.0035) and oral hairy leukoplakia (adjusted OR = 0.67, *p* = 0.033) when compared to MTB uninfected patients. The association of LTBI with a reduced HIV set

study do not fulfill the requirements for open data access: 1) The SHCS informed consent states that sharing data outside the SHCS network is only permitted for specific studies on HIV infection and its complications, and to researchers who have signed an agreement detailing the use of the data and biological samples; and 2) the data is too dense and comprehensive to preserve patient privacy in persons living with HIV. According to the Swiss law, data cannot be shared if data subjects have not agreed or data is too sensitive to share. Investigators with a request for selected data should send a proposal to the respective SHCS address (www.shcs.ch/contact). The provision of data will be considered by the Scientific Board of the SHCS and the study team and is subject to Swiss legal and ethical regulations, and is outlined in a material and data transfer agreement. The numerical data underlying the figures presented in the main manuscript and supplementary information can be found in the files Data_Manuscript.xlsx, Data_AppendixA.xlsx, Data_AppendixB.xlsx.

**Funding:** Swiss National Science Foundation (SNF, http://p3.snf.ch) (grant numbers 33CS30_177499 and 324730B_179571), received by HFG. Yvonne-Jacob Foundation (https://stiftungen.stiftungsschweiz.ch/organizations/stiftung-yvonne-jacob), received by HFG. SNF (grant numbers PZ00P3-142411 and BSSGI0_155851), received by RDK. SNF (grant number P300PB_164742), and grant for scientific development from the University Hospital Zurich, received by JN. SHCS research foundation (Number 857), received by JN. Unrestricted research grant from Gilead Sciences, to the SHCS research foundation. None of the funders had a role in study design, data collection and analysis, decision to publish, or preparation of the manuscript.

**Competing interests:** I have read the journal's policy and the authors of this manuscript have the following competing interests: HFG has received unrestricted research grants from Gilead Sciences; fees for data and safety monitoring board or consulting/advisory board membership from Merck Gilead Sciences, ViiV, Sandoz and Mepha. The institution of HFG received unrestricted educational grants from ViiV, Gilead, MSD, abbvie, Sandoz and Pfizer paid to the institution. JN is supported by a Grant for Scientific Development from the University Hospital Zurich and by Swiss National Science Foundation grant#P300PB_164742. NK received travel grants, lecture fees and fees for advisory board meetings from MSD, Pfizer, and Gilead. Fees for advisory board meetings Janssen, and Basilea

point virus load and fewer unrelated infections in HIV/TB coinfected patients suggests a more complex interaction between LTBI and HIV than previously assumed.

## Background

Models suggest that *Mycobacterium tuberculosis* (MTB) might have emerged as a human pathogen around 400,000 years ago [1]. Over this long period, MTB and humans have evolved to reach a balance; MTB infects many people—approximately 28% of the human population have been exposed to MTB [2]—but over 90% of infected individuals do not develop disease [3]. The evidence of an immune response to MTB in the absence of clinical disease is termed Latent Tuberculosis Infection (LTBI). LTBI represents a spectrum of outcomes, but the differentiation of individuals who harbor viable bacteria from those who have cleared the infection is currently impossible [4,5]. The vast majority of research on LTBI has focused on the aspects of the host–pathogen interface that prevent progression to active pulmonary TB. This framework neglects a basic understanding about how LTBI itself alters human biology.

Recent research in animal models suggests that nonlethal pathogens and commensals provide many benefits to the host [6]. Exposure of pathogen-free laboratory mice to naturally occurring, nonlethal mouse pathogens, for example, has profound effects on the composition of the immune system and confers protection against unrelated pathogens, such as *Listeria monocytogenes* [7]. Chronic Herpes virus infection primes the murine immune system to provide antigen-independent beneficial effects [8]. Contained MTB infection itself protects against MTB rechallenge and heterologous challenges (*L. monocytogenes* and Melanoma metastases) through low-grade cytokinaemia and an augmented innate immune response [9].

In humans, the nonspecific impacts of low-grade infections have not been well studied. The best analogy for self-limiting infections in the human system are live-attenuated vaccines. There is a significant body of evidence suggesting that live-attenuated vaccines may provide additional immune benefits beyond protection against the specific vaccine target [10,11]. Specifically, administration of the TB vaccine bacillus Calmette–Guérin (BCG) or measles vaccines in children reduces overall mortality by more than what would be expected by prevention of these 2 diseases alone [12]. Several ongoing clinical trials will shed light on whether the nonspecific benefits of BCG vaccination can be harnessed to prevent progression of the Severe Acute Respiratory Syndrome Coronavirus 2 (SARS-CoV-2) pandemic [13–17].

Based on these findings, we hypothesize that the continuous interaction between MTB and the host during LTBI benefits the host by augmenting the immune response to other, unrelated pathogens. In particular, we hypothesize that patients with LTBI have fewer opportunistic infections (OIs) and can control HIV better compared to MTB uninfected patients. The extension of this hypothesis predicts that active TB, which is associated with a pronounced inflammatory response and loss of the equilibrium between the host and the pathogen, reflects the breakdown of the protective state seen in LTBI and therefore is associated with more OIs and faster progression of HIV infection. Indeed, the detrimental interaction between HIV and active TB has been extensively described [18].

In this study, we investigated the association of MTB status with HIV disease progression (including both the HIV set point viral load (SPVL) and the occurrence of OIs). By controlling for other major known risk factors of HIV disease progression, we specifically tested for effects associated with having LTBI or active TB disease in people living with HIV. Assessing the association of either LTBI or active TB on HIV SPVL adjusted for CD4 T cell count requires

Pharmaceutica. She is a member of Data safety management board of Allecra Therapeutics SAS. KD's institution has received research funding unrelated to this publication from Gilead and sponsorship to specialist meetings from MSD. NK received grants from the Swiss National Foundation during the conduct of the study which are not related to this study. The institution of EB received fees from Gilead Sciences, MSD, ViiV Healthcare, Pfizer, Abbvie and Sandoz for his participation to Advisory Boards and as travel grants. The authors KK, GSO, KZ, HF, PV, PV, MH and RDK have declared that no competing interests exist.

**Abbreviations:** ART, antiretroviral therapy; BCG, bacillus Calmette–Guérin; CI, confidence interval; HET, heterosexual; IDU, intravenous drug user; IGRA, Interferon-gamma Release Assay; LTBI, latent tuberculosis infection; MSM, men who have sex with men; MTB, *Mycobacterium tuberculosis*; OI, opportunistic infection; OR, odds ratio; SARS-CoV-2, Severe Acute Respiratory Syndrome Coronavirus 2; SHCS, Swiss HIV Cohort Study; SPVL, set point viral load; TB, tuberculosis; TST, tuberculin skin test.

prospective sampling of both viral load and CD4 T cells over years in thousands of patients. The Swiss HIV Cohort Study (SHCS) is in a unique position to study the interaction between LTBI and its host as well as to dissect latent and active TB. Most importantly, information on clinical phenotypes in the SHCS is richly detailed [19]. For example, the high granularity of the longitudinal, clinical data allowed us to investigate patients who developed active TB prior to other OIs in time-to-event analyses.

# Results

## Selection of the study population

We included information from 13,326 tuberculin skin reactivity tests of 10,649 patients and 3,978 Interferon-gamma Release Assay (IGRA) results of 3,623 patients. In total, we analyzed test results from 17,243 different time points of 13,675 patients, with 11,057 (80.1%) patients having only 1 test available (see Fig 1). Of all tests, 1,258 (7.3%) were positive. We removed 187 patients with positive and negative results at different time points, leaving 840 patients with LTBI. Active TB was diagnosed in 770 patients, with 367 cases of extrapulmonary and 546 cases of pulmonary TB (see S1 Text). In total, 13,943 patients were included in our analysis, 12,333 (88.4%) MTB uninfected patients, 840 (6.0%) patients with LTBI, and 770 (5.5%) patients with active TB (see Fig 2).

## Characteristics of the study population (Table 1)

The fraction of male patients was 71.8% for MTB uninfected patients, 65.4% for patients with LTBI, and 65.1% for patients with active TB. The median birth year was 1964 for MTB uninfected patients, 1969 for patients with LTBI, and 1962.5 for patients with active TB. Similarly, the median HIV diagnosis year was 1998 for MTB uninfected patients, 5 years later for patients with LTBI, and 4 years earlier for patients with active TB. Moreover, 70.7% of MTB uninfected patients were from the region of Western Europe, while this was the case in 44.5% of patients with LTBI and 47.5% of patients with active TB. The most frequent HIV risk group for MTB infected patients was heterosexual contacts (LTBI: 48.3%, active TB: 47.8%), and men who have sex with men (MSM) (39.2%) for MTB uninfected patients. The median years of follow-up was 9.6 for MTB uninfected patients, 9.5 years for patients with LTBI, and 6.5 for patients who developed active TB. While 36.1% of patients with active TB died, this was the case in 21.6% for MTB uninfected patients and 9.3% of patients with LTBI. However, many patients were lost to follow-up in all 3 groups (no TB infection: 22.5%, LTBI: 29.5%, active TB: 22.7%), so the actual fraction of patients who died might strongly differ from the confirmed death cases. The first CD4 cell count and the CD4 nadir was lowest for patients with active TB (median first CD4 count: 195.5, median CD4 nadir: 70) and highest for patients with LTBI (median first CD4 count: 455.5, median CD4 nadir: 265). Moreover, 52.1% of MTB uninfected patients and 28.2% of patients with LTBI had at least 1 OI, with a total of 109.8 OI per 1,000 person years in MTB uninfected patients and 44.7 OI per 1,000 person years in patients with LTBI (see Table 1 for more information on basic characteristics of the study population). Both the diagnosis date of OIs as well as the time points of the antiretroviral therapy (ART)-naïve RNA measurements used to calculate HIV SPVL were close to the LTBI test date (Fig 1).

## Association of LTBI and active TB with HIV SPVL

We could determine HIV SPVL values for 4,516 (32.4%) patients (12,512 patients had at least 1 HIV RNA measurement available, 8,616 at least 1 measurement before initiation of ART, and 4,516 of these during chronic infection). Of these, 4,069 (90.1%) were MTB uninfected,

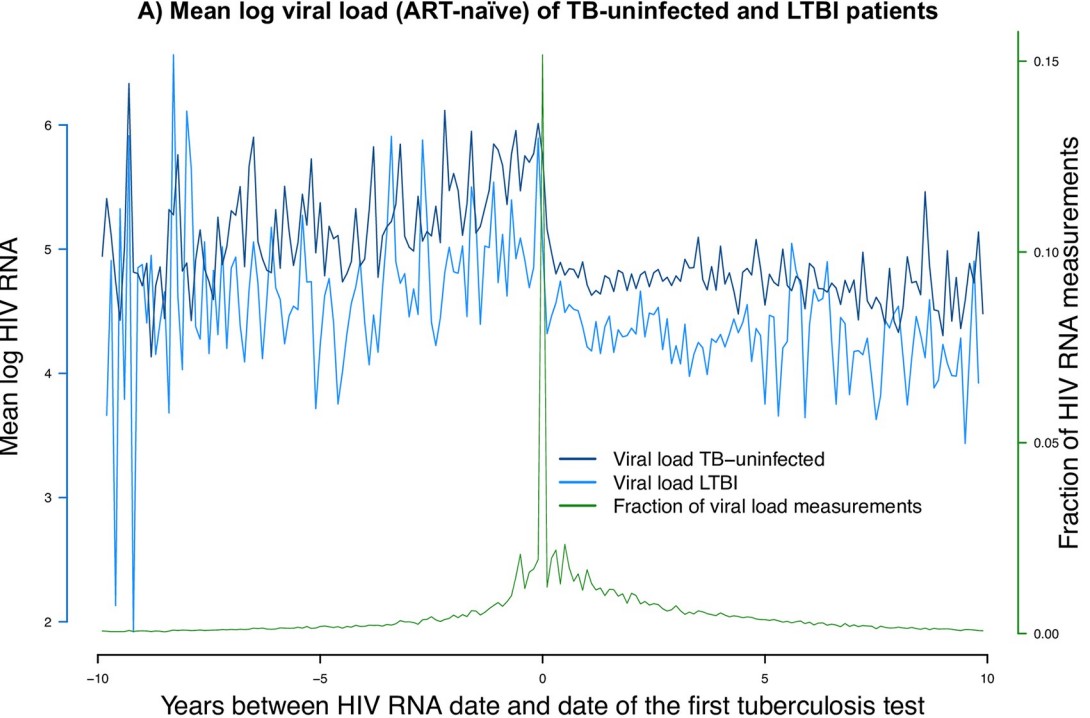

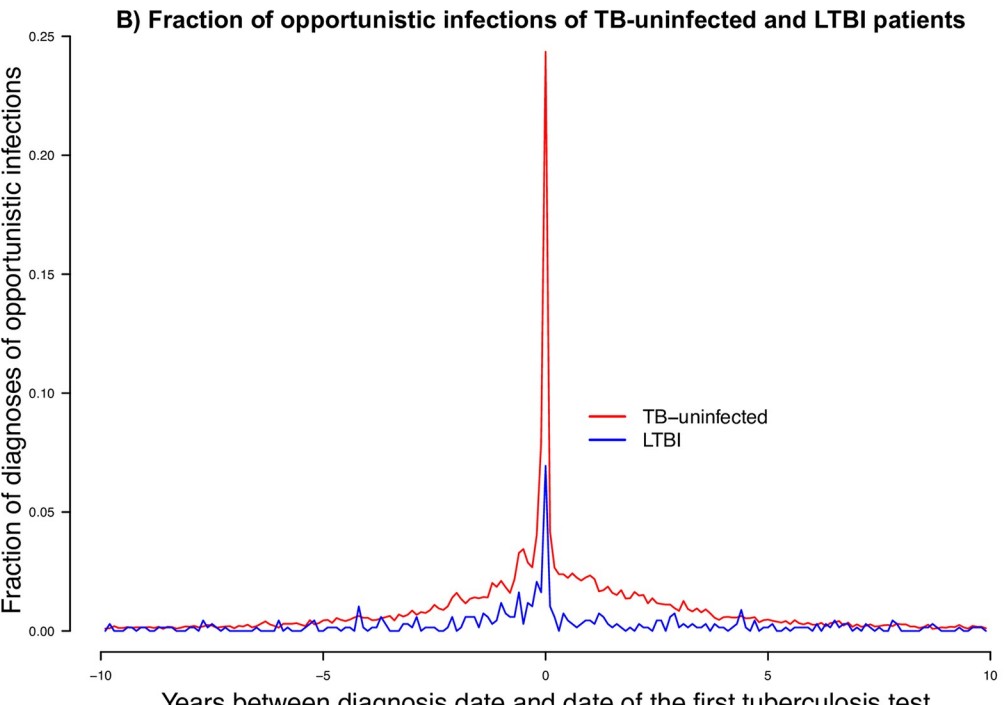

**Fig 1. Timing of the studied events: (A) Mean log viral load measurements of TB-uninfected and LTBI patients and (B) fraction of OIs of TB-uninfected LTBI patients (see S3 Data for the underlying numerical values).** ART, antiretroviral therapy; LTBI, latent tuberculosis infection; OI, opportunistic infection; TB, tuberculosis.

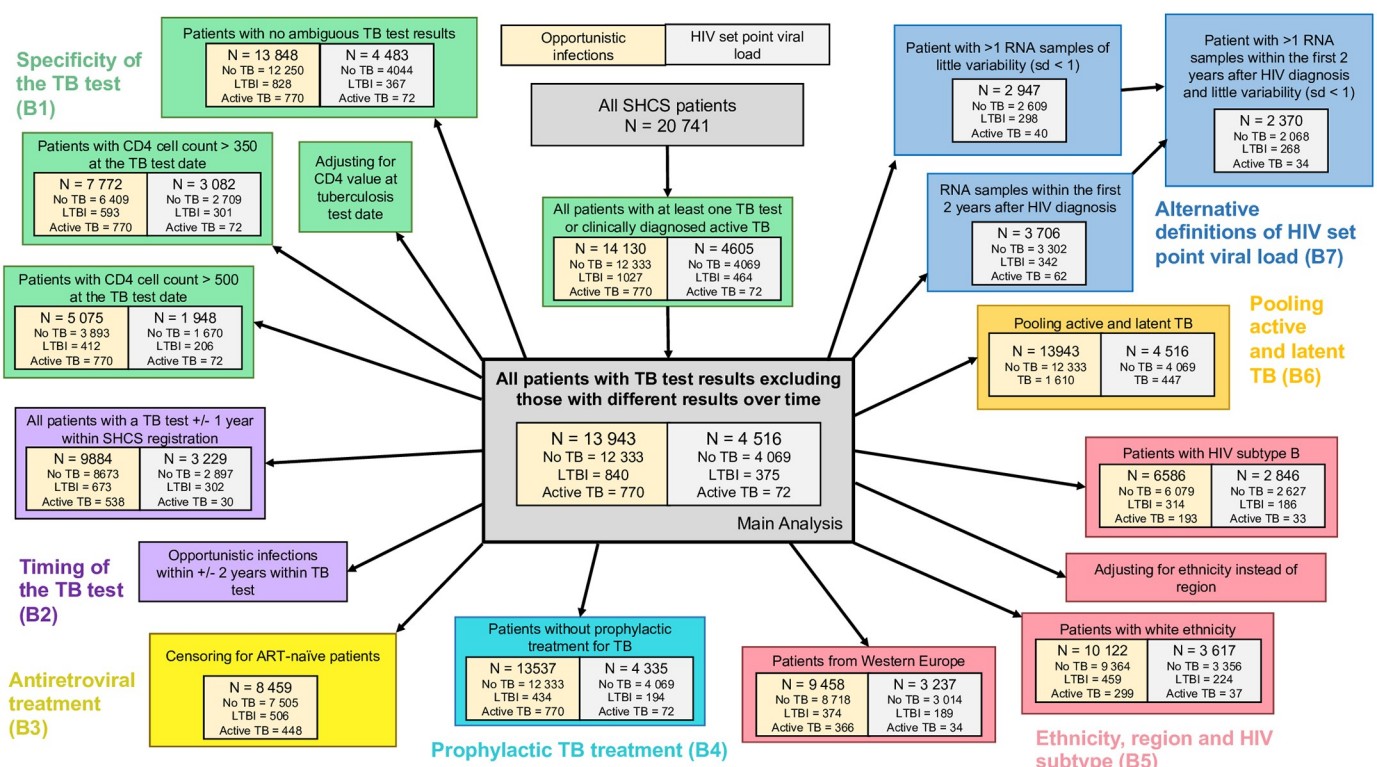

**Fig 2. Description of the study population and the sensitivity analyses; B1 to B7 refer to the respective sections in S2 Text.** ART, antiretroviral therapy; LTBI, latent tuberculosis infection; SHCS, Swiss HIV Cohort Study; TB, tuberculosis.

375 (8.3%) LTBI, and 72 (1.6%) developed active TB. The overall log10 mean HIV SPVL was 4.40 (standard deviation (SD) = 0.75). The log mean HIV SPVL was 4.43 (SD = 0.74) for MTB uninfected patients, 4.11 (SD = 0.71) for patients with LTBI, and 4.63 (SD = 0.80) for patients with active TB (see Fig 3A). In the unadjusted linear regression model, LTBI was associated with a 0.32 (confidence interval (CI) = [0.24, 0.40], $p < 0.0001$) decrease in HIV SPVL (log10 RNA) compared to MTB uninfected patients and with a decrease of 0.21 (CI = [0.13, 0.28], $p < 0.0001$) in the adjusted model (see Fig 3B). This reduction in HIV SPVL remained significant in all sensitivity analyses when restricting the study populations, pooling patients with LTBI and active TB as well as for alternative definitions of HIV SPVL (see Fig 2 and S2 Text). For patients with active TB, we observed an increased HIV SPVL as compared to MTB uninfected patients.

## Association of MTB status with opportunistic infections

The 10 most frequent OIs (excluding pulmonary and extrapulmonary TB) in the study population were candida stomatitis (3,860 cases), oral hairy leukoplakia (1,772 cases), herpes zoster multidermatomal or relapse (1,553 cases), esophageal candidiasis (1,289 cases), *Pneumocystis jiroveci* pneumonia (1,261 cases), HIV-related thrombocytopenia (894 cases), Kaposi sarcoma (632 cases), HIV-related encephalopathy (452 cases), cerebral toxoplasmosis (423 cases), and bacterial pneumonia (396 cases) (see S1 Text). In the unadjusted analysis, all tested OIs were significantly less frequent in patients with LTBI as compared to MTB uninfected patients (see Fig 4). In the adjusted model, LTBI was associated with significantly fewer cases of candida stomatitis (OR = 0.68, CI = [0.52, 0.87], $p = 0.004$) and oral hairy leukoplakia (OR = 0.67,

**Table 1. Basic characteristics of the study populations: MTB uninfected patients, patients with LTBI, and patients with active TB.**

| Variable | | MTB uninfected | LTBI | Active TB |
|---|---|---|---|---|
| Total (*n*) | | **12,333** | **840** | **770** |
| Sex | male (*n*, %) | 8,861 (71.8%) | 549 (65.4%) | 501 (65.1%) |
| Birth year (median, IQR) | | 1964 [1958,1972] | 1969 [1961,1977] | 1962.5 [1957,1970] |
| Ethnicity | white (*n*, %) | 9,364 (75.9%) | 459 (54.6%) | 299 (38.8%) |
| Region | Western Europe (*n*, %) | 8,718 (70.7%) | 374 (44.5%) | 366 (47.5%) |
| HIV subtype | B (*n*, %) | 6,079 (49.3%) | 314 (37.4%) | 193 (25.1%) |
| Diagnosis year (median, IQR) | | 1998 [1990,2007] | 2003 [1997,2009] | 1994 [1987,2002] |
| Registration year (median, IQR) | | 2000 [1994,2009] | 2004 [1998,2011] | 1997 [1990,2005] |
| Transmission group | MSM (*n*, %) | 4,833 (39.2%) | 232 (27.6%) | 143 (18.6%) |
| | HET (*n*, %) | 3,944 (32%) | 406 (48.3%) | 368 (47.8%) |
| | IDU (*n*, %) | 3,020 (24.5%) | 148 (17.6%) | 224 (29.1%) |
| | other (*n*, %) | 536 (4.3%) | 54 (6.4%) | 35 (4.5%) |
| SHCS follow-up | active (*n*, %) | 6,897 (55.9%) | 514 (61.2%) | 317 (41.2%) |
| | lost to follow-up (*n*, %) | 2,775 (22.5%) | 248 (29.5%) | 175 (22.7%) |
| | dead (*n*, %) | 2,661 (21.6%) | 78 (9.3%) | 278 (36.1%) |
| Years of follow-up | total | 138,270.4 | 8,865.3 | 6,919.5 |
| | median, IQR | 9.6 [3.9, 17.7] | 9.5 [3.7, 16.4] | 6.5 [1.9, 15.2] |
| First CD4 count (median, IQR) | | 340 [170,540] | 455.5 [292.8,662] | 195.5 [80,400] |
| CD4 nadir (median, IQR) | | 177 [60,297] | 265 [171.8,385.5] | 70 [20,172.8] |
| Primary infection (*n*, %) | | 815 (6.6%) | 68 (8.1%) | 15 (1.9%) |
| OI | total, at least 1 OI (*n*, %) | 6,467 (52.4%) | 237 (28.2%) | 770 (100%) |
| | all OIs (*n*) | 15,180 | 396 | 2102 |
| | per 1,000 person years | 109.8 | 44.7 | 303.8 |

HET, heterosexuals; IDU, intravenous drug users; IQR, interquartile range; LTBI, latent tuberculosis infection; MSM, men who have sex with men; MTB, *Mycobacterium tuberculosis*; OI, opportunistic infection; SHCS, Swiss HIV Cohort Study; TB, tuberculosis.

CI = [0.46, 0.96], *p* = 0.03). The effects were robust in all sensitivity analyses (see S2 Text for the summary). In stark contrast to the comparison of MTB uninfected patients and patients with LTBI, most tested OIs were more frequent in the unadjusted analysis in patients with active TB compared to MTB uninfected patients. After adjustment, the effect weakened for 8 out of 10 OIs in the case of LTBI and for 4 out of 8 in the case of active TB.

## Time-to-event analysis for candida stomatitis, oral hairy leukoplakia, and herpes zoster

In the time-to-event analysis, LTBI was associated with a lower hazard of candida stomatitis (Fig 5A) as compared to MTB uninfected patients: Without correction for CD4 cell counts, the hazard ratio was 0.33 [0.25, 0.43], after correction 0.48 [0.37, 0.63] (time-updated inclusion of CD4 cell counts as continuous variable) and 0.49 [0.37, 0.64] (inclusion of CD4 cell counts as categorical variable). Independently of MTB status, lower CD4 cell counts were associated with higher hazard ratios of developing candida stomatitis. After additional correction for HIV transmission group and gender, region, and HIV diagnosis year, the hazard ratios were 0.49 [0.37, 0.64], 0.71 [0.54, 0.94], and 0.70 [0.53, 0.92], respectively. Likewise, LTBI was associated with a lower hazard of oral hairy leukoplakia (Fig 5B) when compared to MTB uninfected patients: The hazard ratios in the 3 tested models (no correction for CD4 cell count, time-updated inclusion of CD4 cell counts as continuous variable, inclusion of CD4 cell counts as

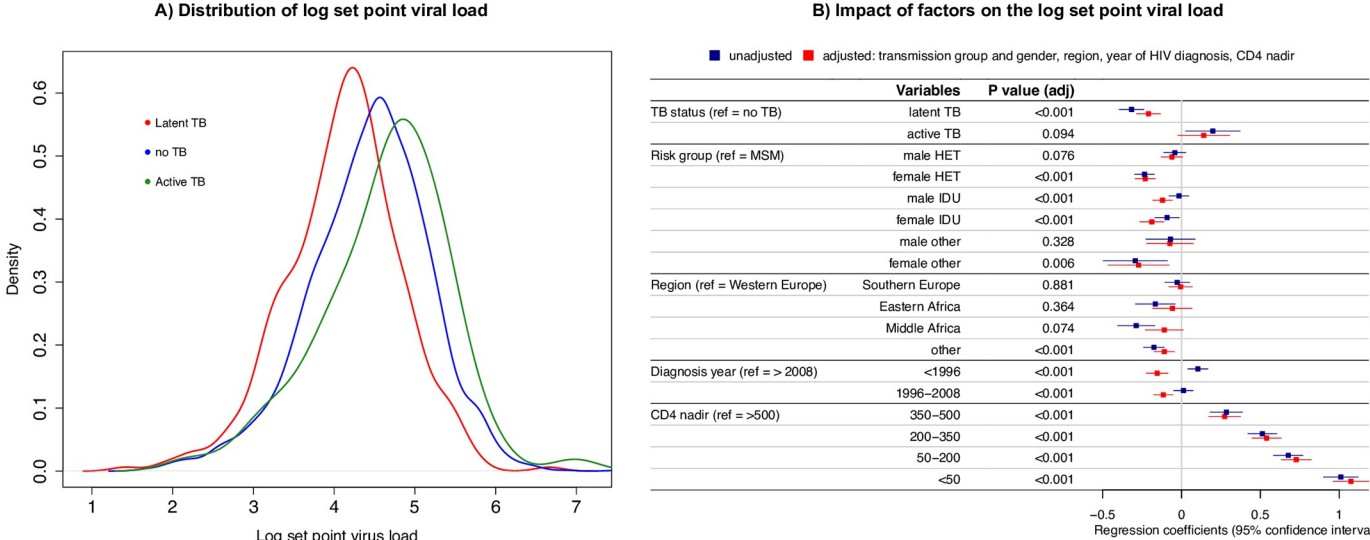

**Fig 3. (A)** Distribution of HIV SPVL values (log RNA) for patients with LTBI, active TB, and MTB uninfected patients. The lines indicated the density function of the log RNA values in the 3 studied groups. **(B)** Association of various factors with the log set point virus load; the lines indicate the 95% CIs obtained in the regression model; the dots indicate the regression coefficients. (see S3 Data for the underlying numerical values) CI, confidence interval; HET, heterosexual; IDU, intravenous drug users; LTBI, latent tuberculosis infection; MSM, men who have sex with men; MTB, *Mycobacterium tuberculosis*; SPVL, set point viral load; TB, tuberculosis.

### Association: Tuberculosis and opportunistic infections

■ unadjusted  ■ adjusted: transmission group and gender, region, year of HIV diagnosis, CD4 nadir

| | Analysis | n,% without TB | n,% with TB | p unadj | p adj | |
|---|---|---|---|---|---|---|
| Candidiasis oral | latent vs no TB | 3429/12333 (27.8%) | 81/840 (9.6%) | < 0.0001 | 0.0035 | |
| | active vs no TB | | 345/770 (44.8%) | < 0.0001 | 0.0001 | |
| Oral hairy leukoplakia | latent vs no TB | 1640/12333 (13.3%) | 34/840 (4.0%) | < 0.0001 | 0.0330 | |
| | active vs no TB | | 103/770 (13.4%) | 0.9244 | 0.0020 | |
| Herpes zoster multidermatomal or relapse | latent vs no TB | 1394/12333 (11.3%) | 48/840 (5.7%) | < 0.0001 | 0.0844 | |
| | active vs no TB | | 106/770 (13.8%) | 0.0412 | 0.3764 | |
| Candidiasis, oesophagial | latent vs no TB | 1110/12333 (9.0%) | 36/840 (4.3%) | < 0.0001 | 0.2736 | |
| | active vs no TB | | 139/770 (18.1%) | < 0.0001 | 0.0033 | |
| Pneumocystis pneumonia | latent vs no TB | 1110/12333 (9.0%) | 25/840 (3.0%) | < 0.0001 | 0.6981 | |
| | active vs no TB | | 131/770 (17.0%) | < 0.0001 | 0.0034 | |
| Thrombocytopenia, HIV-related | latent vs no TB | 802/12333 (6.5%) | 28/840 (3.3%) | 0.0003 | 0.5898 | |
| | active vs no TB | | 61/770 (7.9%) | 0.1313 | 0.6252 | |
| Kaposi sarcoma | latent vs no TB | 567/12333 (4.6%) | 12/840 (1.4%) | < 0.0001 | 0.3849 | |
| | active vs no TB | | 51/770 (6.6%) | 0.0113 | 0.3976 | |
| Encephalopathy, HIV-related | latent vs no TB | 395/12333 (3.2%) | 7/840 (0.8%) | 0.0003 | 0.1358 | |
| | active vs no TB | | 47/770 (6.1%) | < 0.0001 | 0.0679 | |
| Toxoplasmosis, cerebral | latent vs no TB | 345/12333 (2.8%) | 7/840 (0.8%) | 0.0012 | 0.6297 | |
| | active vs no TB | | 67/770 (8.7%) | < 0.0001 | < 0.0001 | |
| Bacterial pneumonia, recurrent | latent vs no TB | 345/12333 (2.8%) | 8/840 (1.0%) | 0.0022 | 0.5409 | |
| | active vs no TB | | 41/770 (5.3%) | < 0.0001 | 0.3372 | |

**Fig 4. Association of the 10 most frequent OIs with TB infection: Patients with active TB and LTBI compared to MTB uninfected patients, respectively (active TB versus no TB, latent versus no TB).** The lines indicate the 95% CIs obtained through the logistic regression model; the dots indicate the ORs. (see S3 Data for the underlying numerical values) CI, confidence interval; LTBI, latent tuberculosis infection; MTB, *Mycobacterium tuberculosis*; OI, opportunistic infection; OR, odds ratio; TB, tuberculosis.

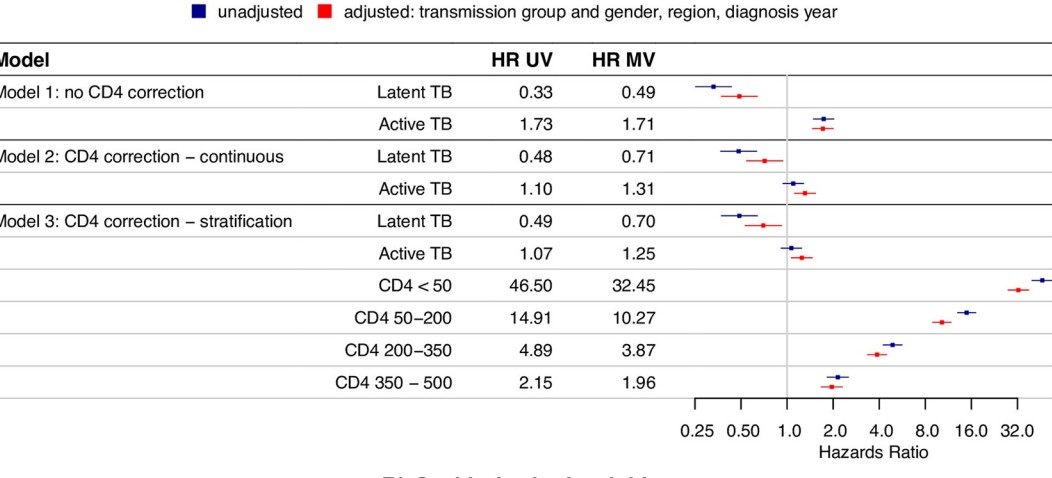

**Fig 5. Time-to-event analysis of the occurrence of candida stomatitis (A), oral hairy leukoplakia (B), and herpes zoster (C): Patients with active TB or LTBI compared to MTB uninfected patients, respectively.** The lines indicate the 95% CIs obtained through the cox proportional hazards model; the dots indicate the HRs. (see S3 Data for the underlying numerical values) CI, confidence interval; HR, hazards ratio; LTBI, latent tuberculosis infection; MTB, *Mycobacterium tuberculosis*; MV, multivariable; TB, tuberculosis; UV, univariable.

categorical variable) were 0.26 [0.17, 0.40], 0.36 [0.23, 0.56], and 0.36 [0.24, 0.56], respectively, and 0.44 [0.28, 0.68], 0.61 [0.39, 0.94], and 0.59 [0.38, 0.92] after additional correction for HIV transmission group and gender, region, and HIV diagnosis year. In the unadjusted analysis of the occurrence of herpes zoster (Fig 5C), we obtained the hazard ratios 0.43 [0.30, 0.62] (no correction for CD4 cell count), 0.55 [0.38, 0.80] (CD4 cell count as continuous variable), and 0.55 [0.39, 0.80] (CD4 cell count as categorical variable). In the adjusted model, we obtained the hazard ratios 0.55 [0.38, 0.80] (no correction for CD4 cell count), 0.70 [0.48, 1.01] (inclusion of CD4 cell counts as continuous variable), and 0.69 [0.48, 1.00] (inclusion of CD4 cell counts as categorical variable). For all 3 tested diseases, the effects only reached borderline significance in some sensitivity analyses when restricting the study population (see S2 Text). Moreover, for all 3 tested OIs, no clear pattern of an association between active TB and OIs was found (see Fig 5). A potential confounder in this cohort could be the fact that active TB is an OI, which potentially might develop prior to the OI of interest, i.e., prior to diagnosed candida stomatitis, oral hairy leukoplakia, or herpes zoster. In a sensitivity analysis, we pooled patients with active TB and LTBI and censored for active TB, i.e., we took into account whether active TB or the OI of interest was first diagnosed. In this sensitivity analysis, hazards were reduced for patients with LTBI for all 3 tested diseases when compared to TB uninfected patients.

## Discussion

In this study, we assessed the association of LTBI infection or active TB with HIV SPVL and the development of OIs at the population level in a prospective, nationwide clinical cohort. Compared to MTB uninfected patients, LTBI was associated with a significant decrease in HIV SPVL, suggesting new and exciting interactions between LTBI and HIV. This effect remained significant after adjusting for HIV transmission group and gender, geographic region of origin, HIV diagnosis year, and CD4 cell counts, and in all sensitivity analyses. In addition, we compared the occurrence of the 10 most frequent OIs (excluding pulmonary and extrapulmonary TB) between MTB uninfected patients, patients with LTBI, and patients with active TB. Compared to 52.4% of MTB uninfected patients, only 28.2% of patients with LTBI developed an OI.

Due to the heterogeneous group of OIs, we analyzed the 10 most frequent OIs separately. In the univariate approach, LTBI was associated with a reduced risk for all tested diseases when compared to MTB uninfected patients. In the adjusted model, the association of LTBI diagnosis with the less frequent occurrence of candida stomatitis, oral hairy leukoplakia, and herpes zoster (the 3 most prevalent OIs) remained significant. The lack of an association between LTBI and the occurrence of other tested OIs could be due to a small number of events, as reflected by similar effects sizes but larger CIs. Ideally, further analyses with a larger sample size within populations of higher MTB prevalence will further dissect the association between LTBI, HIV viral load, and OIs in people living with HIV.

Although the interplay of the altered immune landscape caused by HIV and MTB infection has been studied before, the focus has been almost exclusively on active TB disease. Ongoing HIV replication was shown to be an independent risk factor for active TB [20]. It is well known that decreasing viral load with antiretroviral therapy (ART) lowers the risk for active TB by an order of magnitude prior to the recovery of CD4 T cells [21].

The major strength of our study—the rich and detailed data provided by the SHCS including notifications of OIs and routine MTB testing—allowed us to extend previous studies to interactions between LTBI and HIV. Routine viral load measurements allowed us to determine HIV SPVL of almost one-third of the patients, and routine CD4 cell measurements made it

possible to study the occurrence of OIs in CD4 time-updated models. In addition, the extensive demographic and clinical data in the SHCS allowed numerous multivariate and sensitivity analyses to strengthen our results.

Patients with TB infection (both LTBI and active TB) were less often from Western Europe, were more likely to be infected with non-B HIV subtype, and were female or reported heterosexual contacts as the most likely route of HIV acquisition as compared to MTB uninfected patients. To account for these differences, we included them in the multivariate analyses. We included the cofactor of geographic region in the main analysis instead of ethnicity, as this information was available for almost all patients. To correct for potential differences between these 2 variables (e.g., ethnicity representing host genetic differences and geography representing MTB differences), we adjusted for ethnicity instead of geographic region in a sensitivity analysis. Additionally, we performed independent sensitivity analyses restricting the study population to patients of white ethnicity and HIV subtype B, respectively. Strikingly, throughout these analyses, LTBI decreased the HIV log SPVL as much as well-studied host genetic factors (e.g., the 0.32 log decrease attributed to HLA-C locus polymorphisms [22]).

The lack of a gold standard in tests for defining LTBI is a fundamental limitation to all studies in the field. Since MTB tests (both tuberculin skin test (TST) and IGRA) rely on a T cell memory response, immunosuppression, often causes false negative tests [3], especially in people living with HIV [23]. To control for this, we repeated our analyses for patients with either >350 or >500 CD4 T cells at the time of the MTB test (B1.3). In another analysis, we corrected for the CD4 cell count at the MTB test date (B1.3). The robustness of our results in these analyses suggests that misdiagnosis due to immune suppression plays at most a minor role. To further correct for false negative tests, patients who developed active TB were classified as MTB infected, regardless of MTB test results.

Another difficulty in defining LTBI is the dynamic nature of the course of MTB infection over a lifetime [4,5]. In this study, we assumed "LTBI" is a stable condition since most of the OIs were diagnosed approximately at the same time (Fig 1). The clustering in time is an artifact of clinical care: Patients are often diagnosed with an OI prior to the HIV diagnosis and receive the MTB test soon after HIV diagnosis. ART and prophylactic antibiotic treatment decrease the probability of OIs further. Therefore, we are focusing on clinical observations proximal to HIV diagnosis and MTB testing; for most patients, the duration of this time period falls within the months–years range over which a T cell response is considered stable [4].

Since this is a multicenter study conducted over multiple decades, we are unable to provide details about tuberculin types, the thresholds of IGRAs, etc. While the study design lacks test-level granularity, we benefit from decades of follow-up across multiple centers with close to 14,000 patients. Setting aside the difficulties in LTBI definition, our data indicate that at the time point of HIV diagnosis, the detection of a peripheral MTB-specific T cell response is associated with reduced HIV SPVL and reduced occurrent of the 3 most common OIs in HIV–infected patients.

Categorizing patients based on MTB status introduces a potential bias: Patients with low CD4 cell counts might have died or developed the OI of interest before developing active TB. To account for the time aspect, we performed a sensitivity analysis pooling all MTB-infected patients, censoring for the diagnosis of active TB in the time-to-event analysis (B6). All 3 tested OIs occurred significantly less frequently in MTB-infected patients in this analysis. Moreover, we performed a sensitivity analysis excluding patients who were prophylactically treated for TB (B4): All results remained significant, even though this almost halved the study population.

We included patients with active TB to test the reverse of our primary hypothesis: Once immunological control is lost over asymptomatic LTBI infection, active TB fuels CD4 depletion, weakens the immune system, and increases susceptibility to other diseases [12]. In line

with our hypothesis and in contrast to LTBI, almost all tested diseases occurred more frequently in patients with active TB when compared to MTB uninfected patients prior to adjustment for confounders. CD4 T cell count provides a reliable surrogate of immune competence in people living with HIV [24]. When we included CD4 cell counts as a confounder in an array of multivariate models and sensitivity analyses investigating the effect of active TB on the occurrence of OIs and HIV SPVL, all observed effects either weakened or disappeared.

The lack of a significant effect after adjustment has 2 possible explanations: First, increased frequency of OIs and active TB concurs because of significant immunodeficiency or second, active TB weakens the immune system leading to the occurrence of additional OIs. Either way, the disappearance of the effect after adjustment suggests that our model accounts for the most important confounders. These data agree with the well-documented association between OIs and active TB in people living with HIV and the nonsignificant increase in HIV SPVL for active TB after adjustment observed in a South African cohort [18,25]. However, it cannot be excluded that the results found in then multivariable model are due to residual confounding (most notably for herpes zoster).

The logical extension of the pattern seen in active TB would be that LTBI is associated with lower HIV SPVL and fewer OIs because of changes in CD4 T cell counts. Our findings do not support such a model. In particular, candida stomatitis and oral hairy leukoplakia occurred less frequently in patients with LTBI as compared to MTB uninfected patients after accounting for CD4 cell count and other cofactors in the multivariate model. That the associations in LTBI are independent of CD4 T cells is substantiated by reduced hazards of these infections in a CD4 cell time-updated time-to-event analysis. Since known immunological confounders do not explain the association between LTBI and decreased HIV SPVL and OIs described in this study, we suggest that additional research should address how LTBI alters the host immune response.

As with every observational study, causation of an observation is impossible to prove. We cannot rule out that the beneficial effects observed for patients with LTBI are due to host-specific factors that protect against active TB and other OIs and simultaneously improve control of HIV. Therefore, untested features of the innate immune system (e.g., macrophages [26]) could explain the association between LTBI and lower HIV SPVL. However, this explanation would require that those patients also maintain a more robust MTB-specific CD4 T cell response in peripheral blood.

To summarize, we demonstrate that LTBI was associated with a reduced HIV SPVL and fewer cases of the most prevalent OIs on a population level. These associations were robust to adjusting for the most important demographic and clinical confounders. Independently, various sensitivity analyses further strengthened these observations. These findings support the hypothesis that LTBI can benefit host immune responses and provides new avenues for future research to continue to unravel the complex interactions between mycobacteria and humans.

## Methods

### The Swiss HIV Cohort Study

The SHCS, launched in 1988, is a prospective, multicenter cohort study enrolling adults living with HIV in Switzerland (www.shcs.ch) [19]. The SHCS is a nationwide cohort with 7 centers: Zurich, Basel, Bern, Geneva, Lausanne, Lugano, and St. Gallen. Demographic information and the medical history regarding ART, CD4 cell measurements, HIV RNA, and OIs is collected at study registration. Further clinical and laboratory information is prospectively collected in half yearly follow-up visits.

## Study population and definitions

In the SHCS, 1 TB test is usually performed around study registration, further tests are not standard but all test results are recorded. In line with routine clinical practice and in line with all major clinical guidelines, most of the patients had only 1 LTBI test. In our study, all patients with at least 1 tuberculin skin reactivity test or IGRA for MTB, or clinically diagnosed active TB (see S1 Text), were included in our analysis. In the main analysis, patients with positive and negative MTB tests at different time points were excluded, but included in a sensitivity analysis (see S2 Text). LTBI was defined as a positive skin reactivity test or positive IGRA and no development to active TB during follow-up. The MTB test results were obtained in the form of P (positive), N (negative), and B (borderline) entries, i.e., the interpretation of the test results was performed by the treating physicians. Active TB was defined as at least 1 entry for clinically diagnosed pulmonary or extrapulmonary TB. Most MTB tests were performed around SHCS registration; however, we used all TB test results provided in the SHCS, including tests performed before, during, or after SHCS study entry. In a sensitivity analysis, we restricted our study population to patients with TB tests within 1 year of SHCS registration (see S2 Text). Patients were assigned to the group of MTB uninfected if all MTB tests during the observation period were negative. The stratification into the 3 groups (MTB uninfected, LTBI, or active TB) were fixed throughout time, as most of the study measurements (TB test, diagnosis of OIs, and viral load measurements used for SPVL) clustered around SHCS registration (see Fig 1 and S1 Text).

The HIV diagnosis year was defined using the earliest information available: either a documented positive HIV test or the registration year of the SHCS. HIV risk group was defined as the most likely transmission route: MSM, HET, IDU, or other. Geographic regions of origin were reported according to the UNAIDS region codes. CD4 nadir was defined as the lowest CD4 cell count ever reported in the SHCS. For the calculation of HIV SPVL, only ART-naïve measurements were considered. HIV SPVL was then defined as the mean of all ART-naïve log RNA measurements in the chronic phase of the HIV infection, i.e., at least 90 days after the first positive test and before occurrence of any opportunistic infection.

## Statistical analysis

In the first analysis, the association between MTB status (LTBI, active TB, or TB uninfected) and HIV SPVL was investigated using linear regression, with TB uninfected being the reference group. The model was adjusted for HIV transmission group and gender (MSM, male HET, female HET, male IDU, female IDU, male other, female other—where "other" includes all transmission modes other than MSM, HET, and IDU, as well as unknown transmission mode), geographic region, HIV diagnosis year, and CD4 nadir.

In the second analysis, the association between MTB status and the occurrence of OIs was tested using logistic regression, again with TB uninfected being the reference group. We tested the 10 most frequent OIs diagnosed in the study population, excluding pulmonary or extrapulmonary TB (see S1 Text). Again, the model was adjusted for HIV transmission group and gender, geographic region, HIV diagnosis year, and CD4 nadir. Additionally, we used cox proportional hazard regressions to model the association of MTB status on the hazard of the 3 most frequent OIs. In these cox proportional hazard regressions, 3 different approaches were used to assess the impact of CD4 cell counts: (1) no correction for CD4 cell counts; (2) inclusion of CD4 cell counts in the form of a continuous variable (time-updated for each new value); and (3) inclusion of CD4 cell categories (<50, 50 to 200, 200 to 350, 350 to 500, and >500 cells, time-updated for each new value). In all models, the observation time started with

the first available CD4 measurement until either an event, i.e., diagnosis of the OI of interest, or censoring for death or loss to follow-up. Again, the model was adjusted for HIV risk group and gender, geographic region, and HIV diagnosis year.

### Sensitivity analysis

An overview of all performed sensitivity analyses and the respective study size is illustrated in Fig 2, and details can be found in S2 Text. To understand the potential impact of our definitions of the study population on the observed associations, we performed numerous sensitivity analyses: First, we assessed the impact of different ways of defining MTB infection (B1). For this, (1) we included all patients with at least 1 TB test (including those with different results over time); (2) we excluded patients with ambiguous test results (borderline or positive and negative for the 2 type of tests); (3) we restricted the study population using CD4 cell count at the TB test date in 2 independent analyses: (a) at least 350 CD4 cells/mL; and (b) at least 500 CD4 cells/mL. Second, we assessed the impact of the timing of the TB test by (1) restricting our study population to patients with a TB test within 1 year of SHCS registration; and (2) restricting to patients with OIs diagnosed within 2 years of SHCS registration (B2). Third, we assessed the impact of ART in the analysis of OIs by performing a time-to-event analysis restricted to ART-naïve patients and censoring for the start of ART. Fourth, to assess the impact of prophylactic TB treatment, we excluded 406/840 (48.3%) patients classified as "LTBI" who obtained prophylactic treatment (Rifampicin or Isoniazid) (B4). Fifth, to assess the impact of the geographic region of origin and ethnicity, (1) we restricted to patients with HIV subtype B; and (2) we assessed the impact of ethnicity and region in 3 ways: (a) instead of correcting for the geographic region as done in the main analysis, we corrected for ethnicity, i.e., white, black, or other ethnicities; (b) we restricted the analysis to patients of white ethnicity; and (c) we restricted to patients from Western Europe (B5). Sixth, we assessed the impact of TB categorization by performing analyses on pooled patients with LTBI and active TB (B6): In the corresponding survival analysis, we censored for active TB using the time point 1.5 years before the diagnosis of active TB [27]. Seventh, we assessed the impact of our choice of definition of HIV SPVL by restricting to chronic, ART-naïve samples within the first 2 years after HIV diagnosis and excluding patients with large variability in VL measurements (B7). All analyses were performed with R (version 3.4.4; R Foundation for Statistical Computing, Vienna, Austria).

## Supporting information

**S1 Text. Supporting information 1.**
(PDF)

**S2 Text. Supporting information 2.**
(PDF)

**S1 Data. Underlying numerical values of all figures presented in S1 Text.**
(XLSX)

**S2 Data. Underlying numerical values of all figures presented in S2 Text.**
(XLSX)

**S3 Data. Underlying numerical values of all figures presented in the main manuscript.**
(XLSX)

## Acknowledgments

We thank the patients for participating in the SHCS, the study nurses, and physicians for excellent patient care, A. Scherrer, A. Traytel, for excellent data management, and D. Perraudin and M. Amstutz for administrative assistance.

The members of the SHCS are Anagnostopoulos A, Battegay M, Bernasconi E, Böni J, Braun DL, Bucher HC, Calmy A, Cavassini M, Ciuffi A, Dollenmaier G, Egger M, Elzi L, Fehr J, Fellay J, Furrer H, Fux CA, Günthard H (President of the SHCS), Haerry D (deputy of "Positive Council"), Hasse B, Hirsch HH, Hoffmann M, Hösli I, Huber M, Kahlert CR (Chairman of the Mother and Child Substudy), Kaiser L, Keiser O, Klimkait T, Kouyos RD, Kovari H, Ledergerber B, Martinetti G, Martinez de Tejada B, Marzolini C, Metzner KJ, Müller N, Nicca D, Paioni P, Pantaleo G, Perreau M, Rauch A (Chairman of the Scientific Board), Rudin C, Scherrer AU (Head of Data Centre), Schmid P, Speck R, Stöckle M (Chairman of the Clinical and Laboratory Committee), Tarr P, Trkola A, Vernazza P, Wandeler G, Weber R, and Yerly S. We thank Alan Diercks for stinging criticisms.

## Ethic statement

The SHCS was approved by the local ethical committees of the participating centers: Kantonale Ethikkommission Zürich (KEK-ZH-NR: EK-793); Ethikkommission beider Basel ("Die Ethikkommission beider Basel hat die Dokumente zur Studie zustimmend zur Kenntnis genommen und genehmigt."); Kantonale Ethikkommission Bern (21/88); Comité departmental d'éthique des specialités médicales es de médecine communautarie et de premier recours, Hôpitaux Universitaires de Genève (01–142); Commission cantonale d'éthique de la recherche sur l'être humain, Canton de Vaud (131/01); Comitato etico cantonale, Repubblica e Cantone Ticino (CE 813); Ethikkommission des Kantons St. Gallen (EKSG 12/003), and written informed consent was obtained from all participants.

## Author Contributions

**Conceptualization:** Katharina Kusejko, Huldrych F. Günthard, Roger D. Kouyos, Johannes Nemeth.

**Data curation:** Katharina Kusejko.

**Methodology:** Katharina Kusejko, Gregory S. Olson, Roger D. Kouyos, Johannes Nemeth.

**Project administration:** Johannes Nemeth.

**Resources:** Huldrych F. Günthard, Roger D. Kouyos.

**Software:** Katharina Kusejko.

**Validation:** Johannes Nemeth.

**Visualization:** Katharina Kusejko.

**Writing – original draft:** Katharina Kusejko, Roger D. Kouyos, Johannes Nemeth.

**Writing – review & editing:** Katharina Kusejko, Huldrych F. Günthard, Gregory S. Olson, Kyra Zens, Katharine Darling, Nina Khanna, Hansjakob Furrer, Pauline Vetter, Enos Bernasconi, Pietro Vernazza, Matthias Hoffmann, Roger D. Kouyos, Johannes Nemeth.

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
