## [Editor Report · Decision Letter 0]

14 May 2020

Dear Dr Kusejko, 

Thank you for submitting your manuscript entitled "Diagnosis of latent tuberculosis infection is associated with reduced HIV viral load and lower risk for opportunistic infections in people living with HIV" for consideration as a Research Article by PLOS Biology. I sincerely apologize for the delay in getting you a decision. As I’m sure you can understand, our Academic Editors (and reviewers) currently have very limited availability due to COVID-19 related disruptions, and our editorial team is similarly affected. Please do accept our apologies for the unavoidable delays.

Your manuscript has now been evaluated by the PLOS Biology editorial staff as well as by an Academic Editor with relevant expertise and I am writing to let you know that we would like to send your submission out for external peer review.

Please re-submit your manuscript within two working days, i.e. by May 16 2020 11:59PM.

Kind regards,

Hashi Wijayatilake, PhD,

Managing Editor

PLOS Biology

---

## [Decision Letter · Decision Letter 1]

14 Jul 2020

Dear Katharina,

Thank you very much for submitting your manuscript "Diagnosis of latent tuberculosis infection is associated with reduced HIV viral load and lower risk for opportunistic infections in people living with HIV" for consideration as a Research Article at PLOS Biology, and please accept our apologies for the time it has taken for us to contact you with a decision on your work, which is longer than we aim for. Your manuscript has been evaluated by the PLOS Biology editors, an Academic Editor with relevant expertise, and by four independent reviewers, whse expertise and comments can be found below.

As you will see, although all acknowledge the value of the cohort and of the dataset ad are in principle quite interested in the work, referees 1, 2 and 4 especially raise a number of serious issues that question the conclusiveness of your analyses. For example, the fact that there is set-point viral load data for a minority, which could introduce selection bias, that there seem to be substantial losses to follow-up, that whether/how anti-retroviral therapy was analysed as a covariate is unclear, and concerns about the time sequence of events and therefore the direction of causality. Referees 1 and 4 consider that the exclusion of TB discordant people is a shame, as they could also be informative. Referee 3's report is raises some issues regarding the writing in introduction and discussion. The rest referees’ reports are clear and the remaining issues should be straightforward to address.

In light of these reviews, and the constructive nature of the reports, which suggest various ways of addressing confounders/ascertainment bias, we would be happy to invite a revision of the study that addresses the referee concerns in full. We will not be able to make a decision about publication until we have seen the revised manuscript and your response to the reviewers' comments, which will also be sent for further evaluation by the reviewers.

We expect to receive your revised manuscript within 3 months, but please let us know if the revision process is likely to take longer. 

**IMPORTANT - SUBMITTING YOUR REVISION**

3. Your Supplementary Information file will need to be split into its 7 "chapters", as they are referred to in the main text and will need to be linked to each SI file.

*Re-submission Checklist*

*Published Peer Review*

*PLOS Data Policy*

*Blot and Gel Data Policy*

With best wishes,

Nonia

Nonia Pariente, PhD, 

Editor-in-Chief PLOS Biology

PLOS Biology

REVIEWS:

Reviewer expertise:

Reviewer #1: Tuberculosis

Reviewer #2: HIV epidemiology; TB control measures in HIV-endemic populations

Reviewer #3: TB epidemiology

Reviewer #4: TB and HIV co-infections

Reviewer reports:

Reviewer #1: This interesting study investigates how testing positive for latent M.tuberculosis infection (LTBI) in those co-infected with HIV affects the HIV viral load, and the acquisition of other co-infections/co-morbidities. 

There is an extensive literature on how M.tuberculosis afters the immune system, including reducing antigen presentation, and beneficial T cell function. Thus it may not be surprising that patients with clinical TB might have much more marked alterations of their immune system leading to a change in how HIV is controlled. This Swiss cohort provides a valuable opportunity to investigate these interactions between latent and active TB and HIV viral load and the number of opportunistic infections acquired. It is striking that having studied over 13,000 patients, the HIV nadir was 70 in TB compared to 265 in those with LTBI, and that the rate of opportunistic infections was lower in those with latent Mtb infection and HIV infection compared to those without Mtb infection, although as expected those with TB disease and HIV infection had increased rates of opportunistic infections.

There is an extensive set of supplementary information containing 56 supplementary figures., only a few of which are referred to in the text. 

Methods. The patients from the cohort all had positive or negative tests for LTBI; any patients with inconsistent results were excluded. This is a pity, as there is recent interest in those who convert to LTBI positive and then revert to negative. More information about the finding that inclusion of these individuals did not greatly affect the findings might be included. 

Results. The text should be checked to ensure that when the LTBI group are described as having significant decreases in HIV viral load or opportunistic infections etc, it is clear whether it is the Mtb uninfected HIV infected group that the LTBI group is being compared to, or the TB patients group. 

LTBI is not a clear-cut state but rather a spectrum, as until TB is actually diagnosed, someone with incipient or subclinical TB will be categorised as having LTBI. There will also be individuals who test positive for Mtb infection using TST or an IGRA test, who then revert to negative, perhaps because the mycobacteria have been cleared. Overall, there will be fewer live mycobacteria in LTBI than in active TB disease, and as Mtb has the capacity to modulate immunity, it is perhaps less surprising that therefore patients with TB disease will have a more suppressed immune system and therefore be more susceptible to opportunistic infections. 

Secondly, if these are LTBI patients who have not progressed to active TB, due to the long follow-up of participants in this cohort, they will not be typical of LTBI individuals as a whole, as they are those with the ability to prevent progression to active disease, even when HIV-induced immunosuppression is present. In other words they may be a group protected against opportunistic infections as they have not progressed to TB. Assuming that the status "LTBI" is a stable condition, as discussed on page 13 (Discussion) is therefore a potential limitation of the approach taken. It may also be worth noting that progression from LTBI to active TB is likely to occur in the first 1-2 years after infection in about half of those infected who will progress to TB disease; thus if many of this cohort have been infected before coming to Switzerland, again most of those likely to progress to TB shortly after infection will have done so.

A statement might also be included about whether any of the individuals with LTBI were offered chemoprophylaxis.

Minor comments:

Abstract, third paragraph. HIV infection was associated with a reduced HIV set point. The text should clarify if this is compared to those without Mtb infection, or those who developed active TB.

Abstract line 1, change "were ", to "have been " exposed….Also, page 4, line 4. 

Page 9, line 2, change to "Similarly.."

Reviewer #2: This is an interesting paper which uses data from the Swiss HIV cohort to examine the association between latent TB infection (LTBI) and HIV clinical progression, including set-point viral load (VL) and the incidence of opportunistic infections (OIs). It purports to show that HIV+ patients with LTBI have a lower VL and lower incidence of OIs than HIV+ patients uninfected with TB, while there was evidence that patients with active TB disease have higher VL and incidence of OIs.

Strengths of this paper are the very large sample size, extensive data on potential confounding factors and systematic follow-up. However, there are also some serious weaknesses. First, one of the primary outcomes, set-point VL, was only available in about a third of the patients, so there are serious concerns about potential selection bias. There is also quite a high rate of losses to follow-up (over 20% of patients were lost). ART would obviously be an important covariate but there is no information on how many patients were on ART at different times, or how this was incorporated in the analysis. The time sequence of events is also unclear. It seems that the classification of TB infection status is based on TB tests carried out at different times following diagnosis (and enrolment in the cohort), so some of these tests may have been conducted after the outcomes (VL and OIs) were recorded. This is also a problem for the analysis of active TB. Some more detailed comments are given below.

Major comments:

1. The Abstract reports a reduced viral load as a "hazard ratio". This can't be correct?

2. The main paper needs to give clearer information about the test schedule for TB infection in the cohort. Information on this is given in the Supplementary Material, but understanding the timing of these tests, and the reasons for testing, seem important to interpret the results.

3. Later development of active TB is an exclusion criteria for the LTBI group. This will clearly impose a selection effect on this group so that those with more immune dysfunction are omitted. What implications might this have?

4. There are major concerns around the time sequence of events and therefore the direction of causality. A decline in immune function following infection may lead to the LTBI test becoming negative, especially in those with a high viral load and this might induce an apparent association between LTBI negativity and high viral load. This is a particular problem if LTBI test results are used from after the time the VL measurements are made. Maybe the analysis should be restricted to LTBI results taken within a short period after diagnosis?

5. As noted above, there are similar concerns with the analysis of OIs. In many cases, the LTBI test was done long after the occurrence of the OIs.

6. I would have found it useful to have a diagram of the cohort showing the timing of the measurements used in this analysis.

7. A key point is that ART status does not seem to be taken account of in this study. I assume that ART must have been in common use during the latter part of the cohort follow-up and may have had a major impact on TB infection status and on the outcomes of interest in this analysis.

8. Viral load set-point data were only available in about a third of the patients and this could be a major source of selection bias. Did you look at the characteristics of those with and without VL data, to assess the likely degree of bias?

9. In the analyses of OIs, the adjusted effects are considerably smaller than the crude effects and some of the CIs are close to 1. This raises concerns over residual confounding, for example by features of immune status that are not fully captured by the CD4 count, or due to the limited frequency of CD4 testing.

Minor comments:

10. TB is of course a major OI in its own right, but it is never stated that the analysis of OIs excluded pulmonary or extra-pulmonary TB.

11. In the Methods, there is some text about MTB test results, concerning "skin diameter" and "immune status" that are obscure.

12. There is too much emphasis on arbitrary levels of "statistical significance". For example a statement that an effect "vanished" after adjustment whereas in fact the point estimate was still quite substantially below 1 (but the CI just overlapped 0).

13. The analysis of VL takes account of "nadir CD4 count" but there seems to be no limitation on the timing of that measure, which may have been a long time before or after the VL measurement (or the definition of TB infection status).

14. You conclude that because the adjusted effect for active TB was close to zero that confounding must have been adequately controlled in the analysis for LBTI. That is quite a stretch as patterns of confounding may well differ between these different exposures.

15. The treatment of LBTI as a "stable condition" is questionable given that LBTI results in HIV+ may well depend on the immune status of the individual, which can clearly vary over time.

Reviewer #3: Background

I'm not sure why we need to believe that human fire use sparked the evolution of Mtb!

The language is stilted.

The word commensalism is not appropriate for Mtb, as we now know that the pathogen is likely to be in a constant stand-off with the immune system, rather than being simply a commensal. Low grade infection is also an imperfect representation. And it is not likely to be an interaction only with the innate immune system. 

The authors appear to suggest that active TB progresses HIV of its own right, driving HIV to progress. However, this misrepresents the fact that HIV must drive TB progression, as TB disease is far more common in HIV than HIV negative people in the same population. 

The hypothesis being test could be overtly and clearly stated. 

Methods

These are presented soundly

Results

I think these are clearly presented

Discussion

The concluding statement of a hypothesis that wasn't actually the hypothesis is an unfortunate ending to an otherwise solid discussion.

One consideration that isn't mentioned is that the LTBI positive individuals are a selective cohort in relation to TB, as those members that had progressive primary TB disease are excluded. In contrast, individuals who would have progressive primary disease upon infection are still included in the LTBI negative individuals. As such LTBI positive individuals are enriched for those capable of controlling Mtb. It would be reasonable to assume that they are able to control or repel other infections quite well too, including HIV. 

Tables and figures

These are well presented. 

Reviewer #4: This is a well-written and absorbing manuscript which starts with the exciting hypothesis that acquisition of M.tb infection provides a degree of protection against other conditions in the context of HIV co-infection. The authors marshal helpful data to support this. They then go on to show that there are differences between an M.tb uninfected population (the great majority of the Swiss HIV cohort study group being examined), individuals with latent TB infection and those who develop active TB during follow up. This appears to be related to a reduced HIV set point viral load in the latent TB population. Further, they had lower odds of oral thrush and OHL compared to individuals who were considered to be M.tb-uninfected, or had active TB disease.

There may be something in it, but I am not sure that the authors have fully convinced me with the manuscript as it stands. One major issue is that as far as I can understand, the cohorts are defined by their status (i.e. M.tb-uninfected, M.tb-infected and TB disease) at some point in time, and assessments occur going forward from then. Given that the date of diagnoses varied between the three groups (as would be expected active TB was diagnosed at an earlier year to latent TB), and there appears to be little mention of antiretroviral therapy throughout the manuscript, I am struggling a bit with some of their findings. 

For example in the Methods, the authors state that "patients with positive and negative M.tb tests at different time points were excluded". Does this mean that they did or did not perform routine serial testing for latent TB infection? If they did, were those participants who were initially negative and subsequently tested positive not included in the study? If this is a case this feels like a shame as this population may provide insights into their proposed protection conferred by M.tb (for example were less OI events occurring after M.tb infection compared to before?).

It feels like I am missing something here as this is a clear benefit of an observational cohort, which has the flexibility to asses an individual at different points throughout their health trajectory. 

Other comments (plus some further discussion of the above) are given below as they arise in the manuscript:

1. Methods - Study population and definitions. I would be interested to know the number or proportion of participants who had multiple tests? And also those where there were negative then positive results or vice versa.

2. Methods - The authors mention that they used all TB test results including tests performed before, during or after entry to the Swiss HIV cohort study. Can they provide any information on the proportions of these? This might be important if there were a particularly large group diagnosed prior to the Swiss HIV cohort, as there could be concerns about possibly self-reported or non-observed test results.

3. Methods - The authors are keen on HIV set point viral load. This is certainly a useful measure, but I am concerned that its value diminishes in the active TB population, when so many participants in the HIV cohort have presented with TB which then led to an HIV diagnosis, that only around 10% of cases are used in this analysis in the TB disease group. How do the authors answer this point?

4. Methods - Statistical analysis. As mentioned above it would be useful to know what proportion of subjects have an initial negative skin or blood test for latent TB infection and then became positive. Assuming that this does reflect an immune response against M.tb, and samples were assessed at different time points, one might expect there to be the same effect as the authors report in their cross sectional analysis in specific individuals followed over time. Do they have any evidence to demonstrate this?

5. Methods - Statistical analysis. The authors appear to shy away from mentioning antiretroviral therapy. I might have missed it but to me this is so fundamentally important to HIV care and reduction in opportunistic infections and also in controlling true pathogens such as M.tb, that if feels as if they are trying to bury something. Can the authors reassure me about this?

6. Results - As mentioned earlier I am concerned about the relative proportion of individuals who had latent TB compared to active TB. These are pretty much the same number. I think most clinicians would expect to see at least four times as many participants with latent TB than active disease. I appreciate that people can progress from latent infection to active disease, but then things might get messy with regards to where the participant is categorised i.e. as latent infection or active TB disease. Can the authors explain.

7. Results - The information for HIV set point viral load feels to me a bit shaky. The numbers of subjects where this information was available were quite reduced. Is there any evidence that we are looking at some form of selection bias? Again I would be interested to know how the authors have accounted for the use of antiretroviral therapy for an individual and also between the different groups.

8. The Discussion is readable but feels far too long. It takes up five of the twelve total pages of manuscript in my version. I think this could be significantly reduced with no loss of impact.

9. Tables and figures - Table 1 is quite large - I think the authors could collapse some of the sub groups with no loss of impact.

---

## [Decision Letter · Decision Letter 2]

2 Oct 2020

Dear Dr Kusejko,

Thank you for submitting your revised Research Article entitled "Diagnosis of latent tuberculosis infection is associated with reduced HIV viral load and lower risk for opportunistic infections in people living with HIV" for publication in PLOS Biology. I have now obtained advice from two of the original reviewers and have discussed their comments with the Academic Editor. 

Based on the reviews, I am writing with an accept-in-principle decision, which is conditional on you addressing the final minor points of reviewers 2 and 4, whose reports you will find below my signature at the end of this email. You will also need to comply with all of the reporting and formatting requests that follow here and others that you will receive in a separate email from one of my colleagues.

----

In going through your manuscript, we have noted the following issues that need attention:

1) Please restructure the main text according to PLOS Biology format, i.e. Abstract, Introduction, Results, Discussion and Methods. The figures need to be cited in order, and figure 1 is only cited in the Methods, so would either need to be cited at the beginning of the results (best possibility) or be moved to the last main display item.

2) Please include the specific protocol number(s) that were approved by the ethics committees of the participating institutions, if available.

3) In all figure legends please include statistical information, including how many times measurements were performed (and whether they were technical and biological replicates) and, where applicable, what is being represented (e.g. mean or median, plus confidence interval).

4) Please note that per journal policy, all of the individual quantitative observations that underlie the data summarized in the main and supplementary (i.e. those in the apendices) figures and tables of your paper need to be made available in one of the following forms:

a) Supplementary files (e.g., excel). Please ensure that all data files are uploaded as 'Supporting Information' and are invariably referred to (in the manuscript, figure legends, and the Description field when uploading your files) using the following format verbatim: S1 Data, S2 Data, etc. Multiple panels of a single or even several figures can be included as multiple sheets in one excel file that is saved using exactly the following convention: S1_Data.xlsx (using an underscore).

b) Deposition in a publicly available repository. Please also provide the accession code or a reviewer link so that we may view your data before publication. 

Regardless of the method selected, please ensure that you provide the individual numerical values that underlie the summary data displayed throughout the study (note that we do not require all raw data, just the numerical data underlying the figures and tables), as they are essential for readers to assess your analysis and to reproduce it.

6) Please ensure that you update your Data Statement in the submission system to accurately describe where your data can be found. That is, in addition to the restrictions and instructions to obtaining the raw data that you describe, please also state where the numerical data underlying the figures can be found.

------

We expect to receive your revised manuscript within two weeks. Your revisions should address the specific points made by each reviewer, as well as the points I alluded to above. A member of our team will be in touch shortly with a set of additional requests, to ensure formatting and reporting adheres to our guidelines. As we can't proceed until these requirements are met, your swift response will help prevent delays to publication.

- a cover letter that should detail your responses to any editorial requests, if applicable

*Copyediting*

*Published Peer Review History*

*Early Version*

Sincerely,

Nonia

Nonia Pariente, PhD,

Editor-in-Chief,

npariente@plos.org,

PLOS Biology

-------------

Reviewer remarks:

Reviewer #2: I have now been able to review the revised version of this paper. I am pleased to say that I am satisfied with the changes that have been made as well as the response letter from the authors. 

Just a few minor points to pass on to the authors for their consideration when finalising the manuscript for publication:

1. Figure 1a and 1b: The legend and labelling really need to be improved. In Fig 1a, what are the units of the dates (or rather differences of dates) on the x axis? Are they in years? What are the 3 lines marked on Fig 1a?

2. Foot of page 11. The reference to an "interaction between active TB and OIs": The term "interaction" has a specific statistical meaning which is not what is intended here. The word "association" would be clearer.

3. Foot of page 12. "…could be due to lower sample sizes" - I think they are actually referring to the small "numbers of events" rather than "small sample size".

4. Abstract: I think the 0.24 decrease in viral load actually refers to the log viral load? Could this be clarified?

Reviewer #4: Thanks for the useful revisions. 

Figure 1 is helpful, though I couldn't find legends that explain how the X axis timeline should be interpreted, and also indicated to which patient populations the colours in the plots referred.

---

## [Editor Report · Decision Letter 3]

2 Nov 2020

Dear Dr Kusejko,

On behalf of my colleagues and the Academic Editor, Sarah L. Rowland-Jones, I am pleased to inform you that we will be delighted to publish your Research Article in PLOS Biology. 

PRODUCTION PROCESS

Before publication you will see the copyedited word document (within 5 business days) and a PDF proof shortly after that. The copyeditor will be in touch shortly before sending you the copyedited Word document. We will make some revisions at copyediting stage to conform to our general style, and for clarification. When you receive this version you should check and revise it very carefully, including figures, tables, references, and supporting information, because corrections at the next stage (proofs) will be strictly limited to (1) errors in author names or affiliations, (2) errors of scientific fact that would cause misunderstandings to readers, and (3) printer's (introduced) errors. Please return the copyedited file within 2 business days in order to ensure timely delivery of the PDF proof. 

If you are likely to be away when either this document or the proof is sent, please ensure we have contact information of a second person, as we will need you to respond quickly at each point. Given the disruptions resulting from the ongoing COVID-19 pandemic, there may be delays in the production process. We apologise in advance for any inconvenience caused and will do our best to minimize impact as far as possible.

EARLY VERSION

PRESS 

Kind regards,

Alice Musson

Publishing Editor, 

PLOS Biology

on behalf of

Nonia Pariente, PhD,

Editor-in-Chief

PLOS Biology